# The Complex and Well-Developed Morphological and Histological Structures of the Gastrointestinal Tract of the Plateau Zokor Improve Its Digestive Adaptability to High-Fiber Foods

**DOI:** 10.3390/ani12182447

**Published:** 2022-09-16

**Authors:** Xincheng Cai, Darhan Bao, Guohui Ye, Bin Chu, Zhuangsheng Tang, Rui Hua, Limin Hua

**Affiliations:** College of Grassland Science, Gansu Agricultural University, Key Laboratory of Grassland Ecosystem of the Ministry of Education, Engineering and Technology Research Center for Alpine Rodent Pests Control, National Forestry and Grassland Administration, Lanzhou 730070, China

**Keywords:** plateau zokor, gastrointestinal tract, morphology, histology, high-fiber foods, adaptability

## Abstract

**Simple Summary:**

The gastrointestinal tract (GIT) is the main part of the animal digestive system and the morphological and histological traits of the GIT enable species to perform specific functions that enhance the species’ adaptability to their environments. The plateau zokor (*Eospalax baileyi*) is the subterranean herbivorous rodent in the alpine meadow of the Qinghai-Tibet Plateau. The species spends most of their life in closed, underground tunnel systems with lower levels of oxygen, and mainly forage the plants’ roots. This study used comparative anatomy methods to compare the morphological and histological traits of the GIT of both the plateau zokor and the plateau pika (*Ochotona curzoniae*), a small, fossorial lagomorph which forages aboveground plant parts, in order to clarify the traits of the plateau zokor’s GIT and to understand its adaptations to high-fiber foods. Our results showed that the plateau zokor eats abundant, high-fiber foods in the underground tunnel environments, and the specialized morphology and histological structure of the plateau zokor’s GIT provides a favorable guarantee for it to adapt to the energy pressures of high-fiber digestion in harsh environments.

**Abstract:**

The morphological and histological traits of the gastrointestinal tract (GIT) enable the animal to perform some specific functions that enhance the species’ adaptability to environments. The plateau zokor (*Eospalax baileyi*) is a subterranean rodent that mainly forages on plant roots in the Qinghai-Tibet Plateau, but little is known about the mechanism by which the plateau zokor digests roots that have high fiber contents. In this study, we used comparative anatomy methods to compare the morphological and histological traits of the GIT of both the plateau zokor and the plateau pika (*Ochotona curzoniae*), a small, fossorial lagomorph that forages aboveground plant parts, in order to clarify the traits of the plateau zokor’s GIT and to understand its adaptations to high-fiber foods. The results showed that the foods which plateau zokors eat have a higher fiber content than those which the plateau pikas eat. The plateau zokor has a double-chambered and hemi-glandular stomach (the tubular glands are only in the gastric corpus II, and the gastric fundus is keratinized), whereas the plateau pika has a simple, wholly glandular stomach. The gross morphological indicators (organ index and relative length) of the GIT were significantly lower in the plateau zokor than they were in the plateau pika (*p* < 0.001). However, the thickness of the gastric corpus II mucosal layer and the gastric fundus muscle layer are significantly higher in the plateau zokor than they are in the plateau pika (*p* < 0.001), and the thickness of each layer of intestinal tissue is higher in the plateau zokor than it is in the plateau pika. Additionally, the small intestinal villi also are higher and wider in the plateau zokor than they are in the plateau pika. Our results suggest that instead of adapting to digest the high-fiber diet by expanding the size of the GIT, the plateau zokor has evolved a complex stomach and a well-developed gastrointestinal histological structure, and that these specialized GIT structures are consistent with an optimal energy-economy evolutionary adaptation strategy.

## 1. Introduction

Morphological and histological traits enable species to perform certain functions, and the specific morphological and histological characteristics that they have can enhance the adaptability of a species to the environments [1]. In recent years, many researchers have focused on the adaptability of an animal’s external morphology to the environments that it lives in [2,3], while the adaptability of visceral organ morphology to these environments, especially that of digestive organs, has been overlooked. The gastrointestinal tract (GIT) is the main part of the animal digestive system and plays an irreplaceable role in digesting food, such as obtaining external nutrients, maintaining body-fluid and electrolyte balance, and removing metabolic wastes [4]. Differences in species, habitat and lifestyle can affect the digestive strategy of animals, thus causing adaptive variations in the morphological structure and digestive function of the animal’s GIT [5,6]. Studying the adaptability of gastrointestinal morphology and histology is the foundation for understanding the mechanisms by which animals digest and absorb food and acquire energy. It is also the key to clarifying the interaction between organisms and the environments, and it has important physiological and ecological significance.

The mammalian GIT varies greatly in its morphological structure and function, and this is the result of adaptation during long-term natural selection processes. Rodents are the most energy-consuming and diverse group of small mammals, with about 2000 species around the world, accounting for more than 40% of the total number of mammals, and they reproduce rapidly, and they have wide distribution, and can adapt to almost all environmental conditions [7]. Therefore, they are considered to be the most successful animals as a result of long-term evolutionary processes. Rodents are usually small, herbivorous, fossorial animals. Some rodents even live in dark, anoxic, high-humidity, closed underground tunnels for a long time and forage on plant rhizomes for generations [8]. Due to their lifestyle and the environments of the habitats in which they live, these rodents have long been plagued by pressures on food resources and energy in their habitats. To efficiently use food resources and adapt to the uncertainty of environmental energy supply, the GITs in rodents have evolved diversified, specialized, morphological and histological structures and functional characteristics that match these environmental changes [9,10].

The plateau zokor (*Eospalax baileyi*) belongs to the genus *Eospalax* (Spalacidae, Rodentia), the species which spend most of their life in closed, underground tunnel systems at an altitude of 3200–3800 m, mainly feeding on the belowground rhizomes of forbs [11,12]. The plateau zokor is the dominant herbivorous, small mammal in the alpine meadow of the Qinghai-Tibet Plateau (QTP), and it plays an important role in the material cycles and energy flow of the alpine grassland ecosystem [13,14,15]. In terms of its evolution, the plateau zokor differentiated due to the uplift of the QTP (3.6 million years ago), and it developed morphological, physiological, and behavioral adaptations for foraging and living in the underground tunnel environments of high-altitude areas [16,17,18]. It often faces situations where, due to its unique living environment, there is pressure on food resources and energy balance (for instance, foraging activities are extremely energy-consuming and foods is scarce in underground tunnels). To date, only Wang and Wang [19] have studied the gross morphology of the plateau zokor’s GIT, and the results showed that the morphology of the GIT can make adaptive adjustments to varying degrees with seasonal changes. However, little is known about how the plateau zokor’s GIT (especially in terms of its morphological and histological structure) efficiently uses limited and rough food resources to cope with the energy stress in hostile environments.

Fišer [20] contended that the direction of the morphological changes from surface to subterranean animals is not uniform, and some traits that are associated with life in underground environments regress and even disappear, whereas other traits become hypertrophied and more elaborate. In addition, previous studies have generally maintained that food resources and energy demand can significantly affect the morphological and histological structure of the digestive tract in rodents [21,22], and that differences in diet are the main cause of morphological and histological variations in the digestive tract among species [23,24]. In this study, we therefore selected the plateau zokor as our research object and compared it with the plateau pika (*Ochotona curzoniae*), which has a sympatric distribution with the plateau zokor, but it occupies a different niche in terms of food resources (the plateau pika live in holes in alpine meadows, and mainly move on the ground, and prefer to eat the fresh stems, leaves, and buds of the aboveground plants). We compared the morphological and histological structure of the GIT in these two small mammals by employing anatomical methods and considered the findings in the context of possible long-term adaptation of the plateau zokor’s GIT to the habitat and food resources that are available to the animal in its harsh plateau environment. We also tried to verify two hypotheses: (1) the plateau zokor that lives underground forages more high-fiber foods; (2) to make better use of high-fiber foods, the gastrointestinal morphology and histology of the plateau zokor will have undergone adaptive evolution.

## 2. Materials and Methods

### 2.1. Experimental Animals and Their Habitats

Twenty-two adult plateau zokors (nine males and thirteen females) and twenty-one plateau pikas (eleven males and ten females) were captured using snap-traps in October 2020 in Maqu County (33°50′23″ N, 102°08′48″ E; Gansu Province, China). All of the animals that were sampled in this study were healthy adults. (The body weight of plateau zokors is 335.49 ± 18.78 g, and the body weight of plateau pikas is 155.24 ± 5.41 g). All of the captured animals were adults, and they were trapped during the non-breeding season. In this study, seven plateau zokors (three males and four females) and six plateau pikas (three males and three females) were used to study the histological structure of the GIT, while the rest of the animals were used for gross morphological measurements and the description of the GIT.

The sampling area was located on the eastern edge of the Qinghai-Tibet Plateau with an altitude of 3434 m. The climate is cold and humid, with a large temperature difference between day and night, and the solar radiation level is strong. There are only cold and warm seasons all year, which is typical for a plateau climate. The annual average temperature is 1.2 °C, with the highest temperature being 10 °C in July and the lowest temperature being −10 °C in January. The annual average amount of precipitation is 615.5 mm, which mainly occurs from July to September [25]. The area does not have any absolute frost-free periods over the year, and the plant-growing period is approximately 190 days [26]. The type of grassland is an alpine meadow. The plateau zokor and plateau pika are two of the indigenous small mammals in this area, and they have a sympatric distribution pattern. However, the plateau zokor is a subterranean rodent that feeds perennially on the rhizomes of forbs, such as *Potentilla anserina*, *Polygonum viviparum* and *Taraxacum mongolicum* et al. [27], whereas the plateau pika moves on the ground, preferring to feed on fresh stems and leafy parts of Poaceae and Cyperaceae, such as *Elymus dahuricus*, *Kobresia graminifolia*, and *Poaceae annua* et al. [28].

### 2.2. Experimental Design

In order to clarify the morphological and histological characteristics of the plateau zokor’s GIT and to understand the animal’s adaptations to high-fiber foods in its living environments, we determined the composition of the plant nutrients in the habitat and the stomach contents of the plateau zokor and plateau pika, and described and measured the gross morphology and histological structure of their gastrointestinal tissues to allow a comparison between the belowground plateau zokor and the aboveground plateau pika.

### 2.3. Determination of Plant Nutrient Composition and Stomach Contents

Five 0.5 × 0.5 × 0.3 m (length × width × depth) quadrats were randomly selected in both the plateau zokor’s and plateau pika’s habitats using the grid method [12]. After plants and soil were completely separated, the plants were divided into three categories according to their functional groups: grasses (Poaceae), sedges (Cyperaceae), and forbs. Then, the aboveground and belowground parts of the plants were separated and collected for a nutrient composition analysis in the laboratory. In addition, five plateau zokors and five plateau pikas were dissected to remove their stomach contents for a component analysis in the laboratory (the GIT of these animals also were used for gross morphological measurements, subsequently). For both plant and stomach contents, the following components were determined: crude protein content (CPC), crude fat content (CFC), cellulose content (CEC), hemicellulose content (HEC), and lignin content (LIC). We measured the CPC using the Kjeldahl method [29], the CFC using the Soxhlet extraction method [30], and the CEC, HEC, and LIC by employing visible spectrophotometry using special assay kits from the Beijing Solarbio Technology Company [31,32].

### 2.4. Gross Anatomy of the Experimental Animals

The captured animals were euthanized, their sex was identified, their weight and body length was measured, and then their GITs were dissected and photographed [33]. The GIT was carefully separated into the stomach, small intestine, large intestine, and cecum, and the mesenteric, and the other connective tissues were removed. The organs were flattened to the maximum possible length in their natural state (without stretching) using an anatomical plate containing physiological saline, and the length of each part was measured (0.1 cm). The organs were then cut using scissors, longitudinally, and the contents were removed, rinsed with physiological saline, dried on filter paper, and each part of the GIT was weighed (0.01 g).

To allow for the effects of individual body size on GIT morphology, the gastrointestinal organ weight and length were calculated per body weight as the gastrointestinal organ index and relative length of GIT, respectively [34,35], as follows: 

organ index = organ weight (g)/body weight (g) × 100%;

relative length of GIT = length of GIT (cm)/body weight (g).

All experimental protocols and procedures were approved by the Institutional Animal Care and Usage Committees of the Grassland Science College of Gansu Agricultural University (GSC-IACUC-2020-0015).

### 2.5. Determination of the Gastrointestinal Histology

#### 2.5.1. Microstructure

The GIT of each sample was excised and sectioned into small tissue fragments (5 mm) which included the fundus, corpus I, corpus II, the pylorus of the stomach, duodenum (4 cm from the pylorus), jejunum (one-half distance from the small intestine), ileum (4 cm from the cecum), colon, rectum and the cecum (Figure 1). The tissues were then collected and immediately fixed in 4% paraformaldehyde for 24 h. The fixed tissue fragments were washed, dehydrated and immersed in a clearing agent to enhance their transparency, and then embedded in paraffin wax and sliced into 6 μm-thick sections using a microtome (Leica-2016, Leica, Wetzlar, Germany). Then, the sections were stained with hematoxylin and eosin (H&E), and finally they were sealed with gum [36]. Microphotographs of each section were obtained using a digital section scanner (Pannoramic 250, Danjier, Jinan, China) for the histological analysis and determination.

#### 2.5.2. Ultrastructure

Tissue fragments from the GIT were fixed in a 2.5% glutaraldehyde solution at 4 °C for 24 h. Then, the fragments were post-fixed for 2 h in 1% osmium tetroxide, dehydrated in a graded acetone series, and embedded in epoxy resin. Resin polymerization was completed in an oven at 60 °C for 48 h before making ultra-thin sections (50 nm) using a ultramicrotome (Leica-EM UC7, Leica, Wetzlar, Germany). The ultra-thin sections were mounted on copper grids and stained with uranyl acetate and lead citrate [36]. The analysis and photo documentation were performed using a transmission electron microscope (JEM-1400plus, JEOL, Tokyo, Japan).

### 2.6. Data Analysis

The experimental data were preprocessed using Excel 2016 and then analyzed using the SPSS 19.0 software platform. All the figures and tables were completed using the GraphPad prism 8.0 and Excel 2016. The Kolmogorov-Smirnov test and Levene test were used to determine the normal distribution and homogeneity of the variance of the original data before proceeding to further statistical analysis. A comparison of the interspecific differences in morphology, histology and stomach contents of the GITs between the plateau zokor and the plateau pika was performed with an independent-samples *t*-test, and a comparison of the nutrients in the aboveground and belowground parts of the plants was performed with a paired-sample *t*-test. The data in this paper are expressed as mean ± standard error (Mean ± SE), with *p* < 0.05 indicating that the difference is significant.

## 3. Results

### 3.1. Analysis and Determination of Plant Nutrient Composition in the Habitats of the Plateau Zokor and Plateau Pika and in Animal Stomach Contents

We analyzed the nutrients in the aboveground and belowground parts of the plants from the habitats of the plateau zokor and the plateau pika, respectively. The results showed that there was no significant difference in the CPC, CFC and HEC between the aboveground and underground parts of each functional plant group (Figure 2A,B,E). The CEC of the aboveground parts of the Gramineae plants was significantly higher than that of the underground parts (*p* < 0.05), while there was no significant difference in the CEC between the aboveground and underground parts of Cyperaceae and forbs (Figure 2C). The LIC in the belowground parts of Poaceae, Cyperaceae and forbs were significantly higher than that in the aboveground parts (*p* < 0.01) (Figure 2D).

In addition, an analysis of the stomach contents of plateau zokor and plateau pika (Figure 3) showed that the CEC and LIC in the stomachs were significantly higher in the plateau zokor than they were in the plateau pika (*p* < 0.05, *p* < 0.01), while the CFC was significantly lower in the plateau zokor than it was in the plateau pika (*p* < 0.01). There were no significant differences in the CPC and HEC between the plateau zokor and plateau pika.

### 3.2. Gross Morphology of the GIT

Macroscopically, the GIT morphology of the plateau zokor and plateau pika differed greatly (Figure 4A,B), especially in regard to the stomach of the plateau zokor, which was unique and extremely complex (Figure 4C–E). From the external morphological view of the stomach (Figure 4C), there was a deep constriction between the fundus and corpus of the plateau zokor stomach, which almost divided the stomach into two parts. In addition, the fundus of the stomach revealed obvious folds, and the corpus of the stomach had a unique oval structure (gastric corpus II). Internally (Figure 4E), the plateau zokor’s fundus of the stomach was uneven and extremely rough. There was a brush-like structure at the edge of the elliptical structure tissue of the corpus of the stomach, while the area between the fundus and the elliptical structure of gastric tissue was relatively smooth (gastric corpus I). In contrast, the stomach morphology of the plateau pika was similar to that of many other small mammals, except that the corpus of the stomach had a little fold that was situated on the highest part (Figure 4D–F). The intestinal tract of the plateau zokor was similar to that of the plateau pika, but in the plateau zokor the intestinal tract was thicker, the large intestine was longer, and the upper part of the horn-shaped cecum had an obvious spiral structure (Figure 4A,B).

We also measured the weight and length of the gastrointestinal organs of the two animals and calculated their organ index and the relative length of the GIT (Table 1). The results showed that the organ index of each part of the GIT (stomach, small intestine, large intestine and cecum) and the organ index of the gross GIT in the plateau zokor were extremely significantly lower than those of the plateau pika (*p* < 0.001) (Figure 5A, Table 1). The differences in the relative length of each gastrointestinal organ and the gross GIT between the plateau zokor and plateau pika followed the same trends as the organ index did (Figure 5B, Table 1).

### 3.3. Histological Structure of the GIT

The entire GIT of the plateau zokor and plateau pika was essentially a smooth muscle-enveloped tube with an innermost mucosa (including the epithelium, lamina propria and muscularis mucosae), submucosa, muscularis, and a variable outer layer of serosa (Figure 6). The stomach of the plateau zokor differed greatly from that of the plateau pika. The mucosal layer of the plateau zokor’s gastric fundus was covered by a stratified squamous epithelium with a keratinized surface and a developed spongy structure. The lamina propria was thin and without glands. The connective tissue of the submucosa was loose with abundant blood vessels, lymphatic vessels and nerves, and the muscular layer (the muscularis) was thick (Figure 6A and Figure 7A). The histological structure of the gastric corpus I and the pylorus of the stomach was similar to that of the gastric fundus, but they had less surface keratinization and their muscle layer was thicker (Figure 6B–D and Figure 7B–D). The mucosal layer of the gastric corpus II was covered with a single layer of columnar epithelium. Tubular gastric glands were observed in the lamina propria, and these were arranged neatly. The superficial layer of the gastric glands mainly comprised parietal cells. The chief cells were mostly seen at the bottom of the glands, which were conical or columnar. Blood vessels and nerves were seen in the loose connective tissue of the submucosa, and a few lymphocytes and eosinophils were scattered in the stroma. However, the muscle layer of the gastric corpus II was very thin (Figure 6C and Figure 7C). The stomach of the plateau pika was simple. Furthermore, a histological examination confirmed that the mucosal stomach surface revealed rugae, and the stomach was completely glandular (Figure 6A–D and Figure 7A–D).

The characteristics of the small intestinal histological structure of the plateau zokor and plateau pika were similar (Figure 6E–G and Figure 7E–G). The villi of the small intestine were dense, thin, and long, with a finger-like or leafy shape. The mucosal surface of the intestinal villus was covered with a single layer of columnar epithelial cells, with some goblet cells between the epithelial cells (with fewer in the duodenum and more in the ileum). The tubular intestinal glands could be seen in the lamina propria of the intestinal mucosa, and the cells in the connective tissue were closely arranged. Among them, a small number of lymphocytes and macrophages were occasionally scattered in the lamina propria of the duodenal mucosa. The connective tissue of the submucosa was loose, with the presence of blood vessels and nerves. The muscularis layer consisted of smooth muscle which was divided into inner circular and outer longitudinal layers. The serosa layer consisted of both connective tissue and mesothelium. The histological structure of the large intestine was the same as that of the small intestine (Figure 6H–J and Figure 7H–J). However, the intestinal villi were gradually transformed into the distinct folds of the mucosal layer. In addition, the glands of the large intestine in the lamina propria of the mucosa were densely arranged, and a large number of goblet cells were distributed in the glandular epithelium.

We measured the thickness of the mucosa, submucosa and muscular layer of the four parts of the stomach of the plateau zokor and compared them with the plateau pika. The results showed that the mucosal thickness of the gastric corpus II was significantly larger in the plateau zokor than it was in the plateau pika (*p* < 0.001), while the mucosal thickness of other parts of the stomach was significantly smaller in the plateau zokor than it was in the plateau pika (*p* < 0.001). The submucosal thickness of the gastric fundus was significantly smeller in the plateau zokor than it was in the plateau pika (*p* < 0.001), while the submucosal thickness of other corresponding parts of the two animals had no significant interspecific differences. The thickness of the muscle layer in parts of the plateau zokor was larger than it was in the plateau pika, except for the muscle layer of the gastric corpus II, which was significantly smaller in the plateau zokor than it was in the plateau pika (*p* < 0.001), and the gastric fundus muscle layer, which was significantly larger in the plateau zokor than it was in the plateau pika (*p* < 0.001) (Figure 8A–D).

In the histological determination of various layers of the intestinal tract (Figure 8E–J), the thickness of the duodenal mucosa, jejunum mucosa, rectum mucosa and cecum mucosa of the plateau zokor were larger than those of the plateau pika, and there were significant differences in the rectum and cecum (*p* < 0.05), except for the thickness of the colonic mucosa, which was significantly smaller in the plateau zokor than it was in the plateau pika (*p* < 0.001). The submucosal thickness of other intestinal organs was significantly larger in the plateau zokor than it was in the plateau pika (*p* < 0.05), but there was no significant difference in the cecum. The thickness of the muscular layer of the duodenum and cecum was significantly larger in the plateau zokor than in the plateau pika (*p* < 0.05), while the thickness of rectal muscle was significantly smaller than it was in the plateau pika (*p* < 0.05), and there was no significant difference in the other parts. A comparison of the height and width of the intestinal villi showed that the villus height in the duodenum, jejunum, and ileum was taller in the plateau zokor than it was in the plateau pika (Figure 8E–G). The villus width of each part of the small intestine was also greater in the plateau zokor than it was in the plateau pika, and this difference was significant for the duodenum and jejunum (*p* < 0.05).

To clarify the cell composition and cell structure in the gastrointestinal tissues of the plateau zokor, we examined their ultrastructure as these were revealed by the employment of electron microscopy. This showed that the histological structures of the gastric fundus, gastric corpus I, and gastric pylorus of the plateau zokor were similar. The mucosal epithelium was composed of the corneum, stratum spinosum, submucosal lamina propria, and muscularis, as can be seen from the top to the bottom of Figure 9A–C. The mucosal layer was composed mainly of spinous cells (Figure 9D). The spinous processes of the adjacent cells were embedded and connected by desmosomes, and the cytoplasm was rich in microfilaments. There were abundant collagen fibers in the submucosal lamina propria, and scattered capillaries (Figure 9E) and fibroblasts (Figure 9F) were observed between the collagen fibers. The muscular layer consisted of a large number of smooth muscle cells (Figure 9G) and collagen fibers. The electron microscopic examination of the gastric corpus II revealed that the chief cells (Figure 9I), parietal cells (Figure 9J) and the mucous cells (Figure 9K) of the gastric glands (which were branched and tubular, densely arranged with few interstitial components, and occasionally a small number of smooth muscle cells) were found in this part. The chief cells contained a large amount of rough endoplasmic reticulum. There were some well-developed microtubule vesicles and secondary lysosomes in the cytoplasm of the parietal cells. The cytoplasm of the mucous cells contained a large number of mucin particles, and most of them were located above the nucleus. A small number of lipid droplets were also present in the cytoplasm.

The electron microscopic examination showed that the mucosal epithelium of the intestinal mucosa was monolayered and columnar, and this was mainly composed of epithelial cells and goblet cells (Figure 9L–W). The epithelial cells were mostly columnar with an ovular nucleus, and the chromatin was evenly distributed and dominated by euchromatin. There were abundant organelles such as mitochondria, rough endoplasmic reticulum, and ribosomes in the cytoplasm. There were also abundant microvilli on the free surface of the cells (the microvilli were more developed in the duodenum and ileum). The cytoplasm of the goblet cells contained a large amount of rough endoplasmic reticulum and abundant secretory granules, and there was a tightly connected structure on the top of the adjacent cells. The lamina propria of the intestinal mucosa was mostly composed of collagen fibers, with fibroblasts, smooth muscle cells, and blood vessels that were distributed among the collagen fibers (Figure 9X).

## 4. Discussion

The GIT, as the main location for food digestion and nutrient absorption in rodents, plays an important role in the adaptation of animals to increased energy consumption or decreased food quality [5]. In addition, the morphological and histological structures of the GIT are closely related to the animals’ habitat environments, food resources, energy needs, and other factors [34,37]. In particular, dietary differences have been found to be the main cause for the adaptive variations of gastrointestinal morphology and histology between species in small mammals like rodents [24]. In this study, we dissected the GIT of the plateau zokor and plateau pika for morphological and histological examination. We found that the plateau zokor mostly fed on high-fiber foods and had a more complex stomach morphology and well-developed gastrointestinal histological structures, which was compared with the plateau pika which had a sympatric distribution but occupied a better niche for food resources. Therefore, we speculate that the complex and developed morphological and histological structures of the GIT in the plateau zokor contribute to improving the digestibility of high-fiber foods.

Feeding habits are the result of long-term evolutionary processes, with animals gradually changing to suit their habitat through the process of natural selection [38]. It has been extensively demonstrated that differences in diets are often manifested in the morphological and histological structure of the GIT, and the development of different parts of the GIT also reflects an adaptation to food resources [39,40]. The plateau zokor mainly feeds on plant rhizomes because of its tendency of living underground all year round. We analyzed and determined the nutrient components of the belowground parts of the plants in the habitat of the plateau zokor. The results showed that the CEC and LIC in the belowground parts of the forbs plants that the plateau zokor preferred to eat were higher, and the LIC was significantly higher than it was in the aboveground parts of the plants. This was also mirrored in the components of the plateau zokor’s gastric contents (they had a higher CEC and LIC than those in the plateau pika). Therefore, our results fully indicated that the plateau zokor feeds on rough plants that are especially rich in fibers. Valle et al. [41] suggested that rodents can significantly increase the size of their GIT during energy acquisition to compensate for the decrease in digestibility that is caused by low-quality diets. Heroldova and Janova [42] also confirmed this and found that the GIT of the herbivorous vole (*Microtus arvalis*) was relatively larger and more complex than that of the granivorous pygmy field mouse (*Apodemus uralensis*). Based on the analysis of the diets of the plateau zokor, we believe that its GIT may also have undergone long-term adaptive evolution to ensure that the plateau zokor can meet its energy demands for survival and reproduction in the face of poor food resources and the high energy-consuming environments of underground tunnels. 

Certain morphological characteristics of the GIT may enable the digestion and absorption functions to match an animal’s energy needs in order for it to cope with its habitat environments [37,43]. The plateau zokors use high-fiber food resources and have a high energy-consuming lifestyle in the subterranean environments, and their GIT size and weight should therefore be larger than that of their aboveground counterparts, according to the view of Gross et al. [5], Eto et al. [34] and Langer and Clauss [24]. Surprisingly, however, we found that the organ index and relative length of the gastrointestinal organs of the plateau zokor were significantly smaller than those of the plateau pika. Although this result may present interspecific differences in the phenotypic morphology of the organs between different species, we still suggest that caution should be exercised when using the organ index and the relative length of the GIT to measure the adaptability of animals to food resources and energy demands, and that the gross morphology indicators of GIT may not reflect this adaptability well.

Compared with the simple, wholly glandular stomach of the plateau pika, we found the plateau zokor to have a complex, double-chambered, hemi-glandular stomach, in which the mucosal layer was thicker, and the tubular gastric glands existed only in the corpus II, while the mucosal layer of the other parts was a stratified squamous epithelium with a keratinized surface that was without glands, and the muscularis was thicker also. The bilocular stomach anatomy divided the stomach into two separate digestive environments, which is an arrangement in herbivorous rodents that is more conducive to digesting and absorbing high-fiber foods [44]. Kohl et al. [45] suggested that the stomach segmentation creates an environment in the proximal stomach that is suitable for microbial growth. Kohl et al. [46] also found that the foregut microbial communities of the woodrat (*Neotoma* spp.) had a similar density and volatile fatty acid concentrations to those that were found in rumen ecosystems, and the microbiota of the proximal stomach may provide numerous physiological services to the host, such as the initial digestion of fiber, the recycling of endogenous nitrogen and the detoxification of dietary toxins. The hind-stomach contains a large number of glands, and the food mass is digested further by gastric acid and pepsin which are secreted from the glands after the preliminary treatment in the fore-stomach [42]. The plateau zokor feeds mainly on high-fiber plant rhizomes and its stomach contents are rich in cellulose and lignin. We suggest, therefore, that the double-chambered, hemi-glandular stomach structure provides the possibility for the plateau zokor to digest and utilize high-fiber foods for its nutrition.

Our examination of the thickness of each tissue layer in the stomach of the plateau zokor and plateau pika has indicated that the morphological adaptations to a high-fiber diet extend beyond the gross morphology of the stomach. Our results have showed that the mucosal layer of the gastric corpus II was significantly thicker in the plateau zokor than it was in the plateau pika, and the muscular layer in other parts of the stomach was also significantly thicker in the plateau zokor than in the plateau pika. These results indicate that the unique stomach histology of the plateau zokor can not only ensure that the gastric glands secrete enough gastric acid, mucus and other substances for chemical digestion, but—as a result of the thicker muscular layer in the gastric fundus—also strengthen the contraction and peristalsis of the stomach [47], which has a strong mechanical digestion function. Furthermore, our analysis of the gastric ultrastructure of the plateau zokor confirmed that the functional cells in the gastric tissue were highly differentiated and highly matched with their high-fiber food, indicating that they have a strong digestive ability. Therefore, we infer that the double-chambered, hemi-glandular stomach of the plateau zokor is highly adapted to the food compositions of its living environments, and that the food that is available in this environment has driven the evolution of the stomach to become more specialized and efficient in digesting high-fiber foods.

The intestinal tract of the plateau zokor also appeared to be suited to the animal’s environments. The intestinal tract is the main location for food re-digestion and nutrient absorption in animals [48]. It can effectively adapt to variations in energy balance and available foods by changing its morphological and histological structure [34]. A developed intestinal tissue structure can be especially effective at improving an animal’s digestion and absorption of nutrients [49]. The results of this study showed that the intestinal histological characteristics of the plateau zokor and plateau pika were similar, but the intestinal histological structure of the plateau zokor was developed and had more digestive advantages, and the villi length, width and thickness of each tissue layer were larger in the plateau zokor than they were in the plateau pika. Compared with the surface environments of the plateau pika, the underground tunnels that are inhabited by the plateau zokor severely restrict the quality and availability of food for the plateau zokor. We surmise that the developed intestinal tissue structure of the plateau zokor is the result of long-term adaptation by the animal to its underground environments. The extent of digestive adaptions was also apparent under the electron microscope, where the intestinal tract of the plateau zokor was observed to be rich in intestinal gland cells, goblet cells and autophagy cells, with abundant microvilli, a large number of mitochondria and a large amount of rough endoplasmic reticulum. Those findings indicate that the intestinal tissue of the plateau zokor has obvious digestion advantages, and that they can maximize the absorption of limited food to ensure the energy supply for the plateau zokor to survive in its harsh underground environments.

## 5. Conclusions

In summary, our results have showed that the plateau zokor eats abundant, high-fiber food in the underground tunnel environments, and the specialized morphology and histological structure of the plateau zokor’s GIT provide a favorable guarantee for it to adapt to the energy pressures and high-fiber food digestion in harsh environments. Instead of adapting to the high-fiber diets by expanding the size of the GIT in an environment with an uncertain energy supply, the plateau zokor has evolved a complex, double-chambered, hemi-glandular stomach and a well-developed gastrointestinal histological structure (with, for example, an increase in the thickness of the tissue layer and an increase in the length and width of the intestinal villi), which is consistent with an optimal energy–economy evolutionary adaptation strategy, since the evolution of a larger GIT would be an extremely energy-demanding strategy for the plateau zokor.

Based on our research results, we believe that small mammals may preferentially change the histological structure, rather than rapidly change the size, of their GIT when facing food deterioration or increased energy demands in their habitat over a long period. However, our study was limited to the morphology and histology of the GIT. The adaptive evolution of cell function, enzyme activity and the microbial community in the GIT is another important means by which animals may enhance their digestive function and this also merits further study.

## Figures and Tables

**Figure 1 animals-12-02447-f001:**
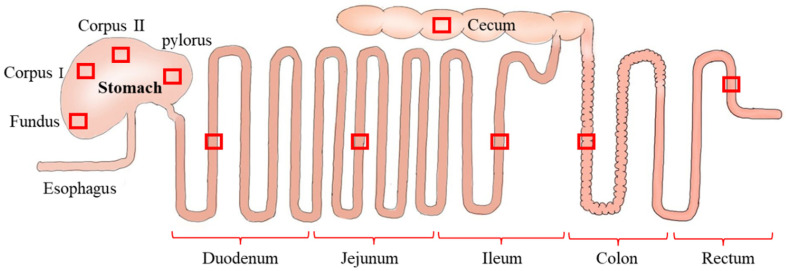
Schematic diagram of gastrointestinal organs of small mammals. The site of tissue harvesting is indicated by red squares for histological observation and measurement of the GIT.

**Figure 2 animals-12-02447-f002:**
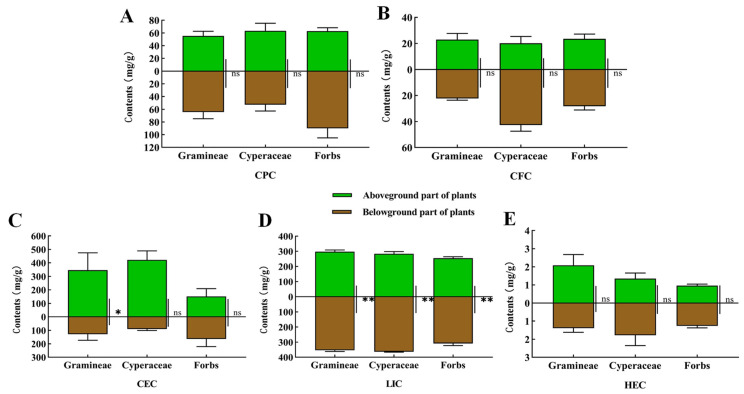
Plant nutrients in the habitats of the plateau zokor and plateau pika. (**A**) The CPC of plant nutrients; (**B**) The CFC of plant nutrients; (**C**) The CEC of plant nutrients; (**D**) The LIC of plant nutrients; (**E**) The HEC of plant nutrients. * indicates significant difference (*p* < 0.05); ** indicates an extremely significant difference (*p* < 0.01); ns indicates an insignificant difference.

**Figure 3 animals-12-02447-f003:**
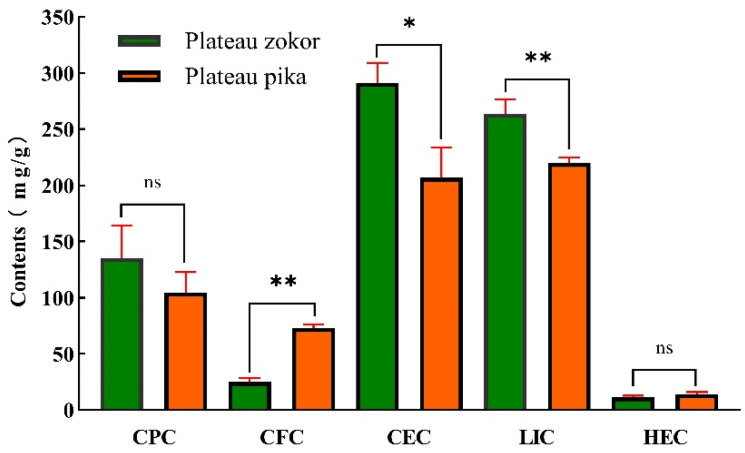
Differences in stomach contents between the plateau zokor and plateau pika. * indicates significant difference (*p* < 0.05); ** indicates an extremely significant difference (*p* < 0.01); ns indicates an insignificant difference.

**Figure 4 animals-12-02447-f004:**
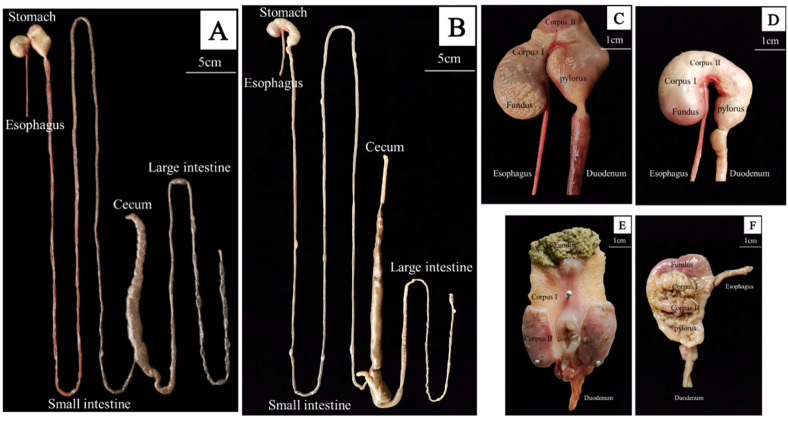
Gross morphological view of the gastrointestinal organs of the plateau zokor and plateau pika. (**A**) Gross morphological view of the GIT of the plateau zokor; (**B**) gross morphological view of the GIT of the Plateau Pika; (**C**) external morphological view of the stomach of the plateau zokor; (**D**) external morphological view of the stomach of the plateau pika; (**E**) internal morphological view of the stomach of the plateau zokor; (**F**) internal morphological view of the stomach of the plateau pika.

**Figure 5 animals-12-02447-f005:**
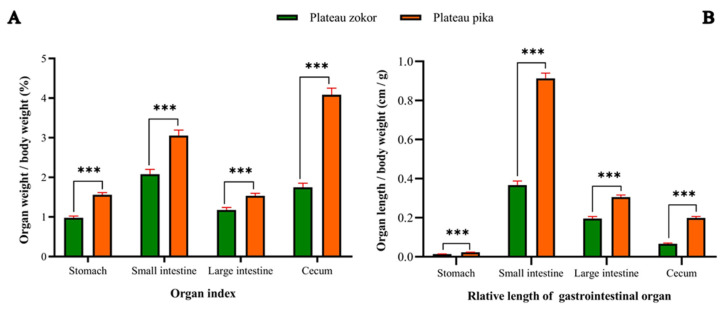
Differences in gastrointestinal organ index and relative length between the plateau zokor and plateau pika. (**A**) Differences in gastrointestinal organ index between the plateau zokor and plateau pika. (**B**) Differences in relative length between the plateau zokor and plateau pika. *** indicates an extremely significant difference (*p* < 0.001).

**Figure 6 animals-12-02447-f006:**
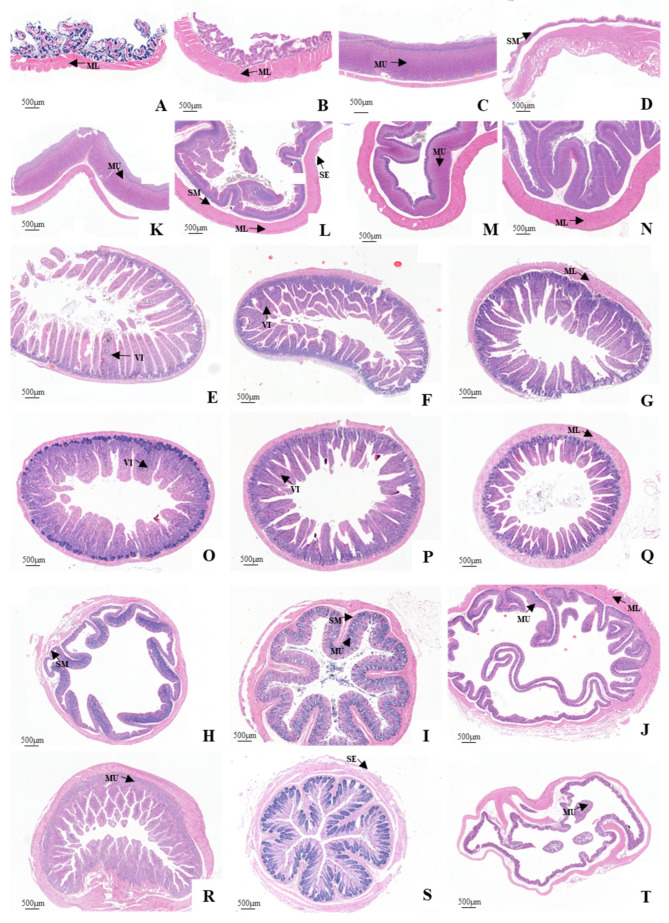
Comparison of the panoramic images of the tissue sections of the GIT of the plateau zokor and plateau pika (40× magnification). (**A**–**J**) Stomach, ((**A**–**D**) fundus, corpus I, corpus II, and pylorus of stomach), duodenum, jejunum, ileum, colon, rectum, and cecum of the plateau zokor, respectively; (**K**–**T**) gastrointestinal organs of the plateau pika in the same order as above. ML: muscularis; MU: mucosa; SE: serosa; SM: submucosa; VI: villi.

**Figure 7 animals-12-02447-f007:**
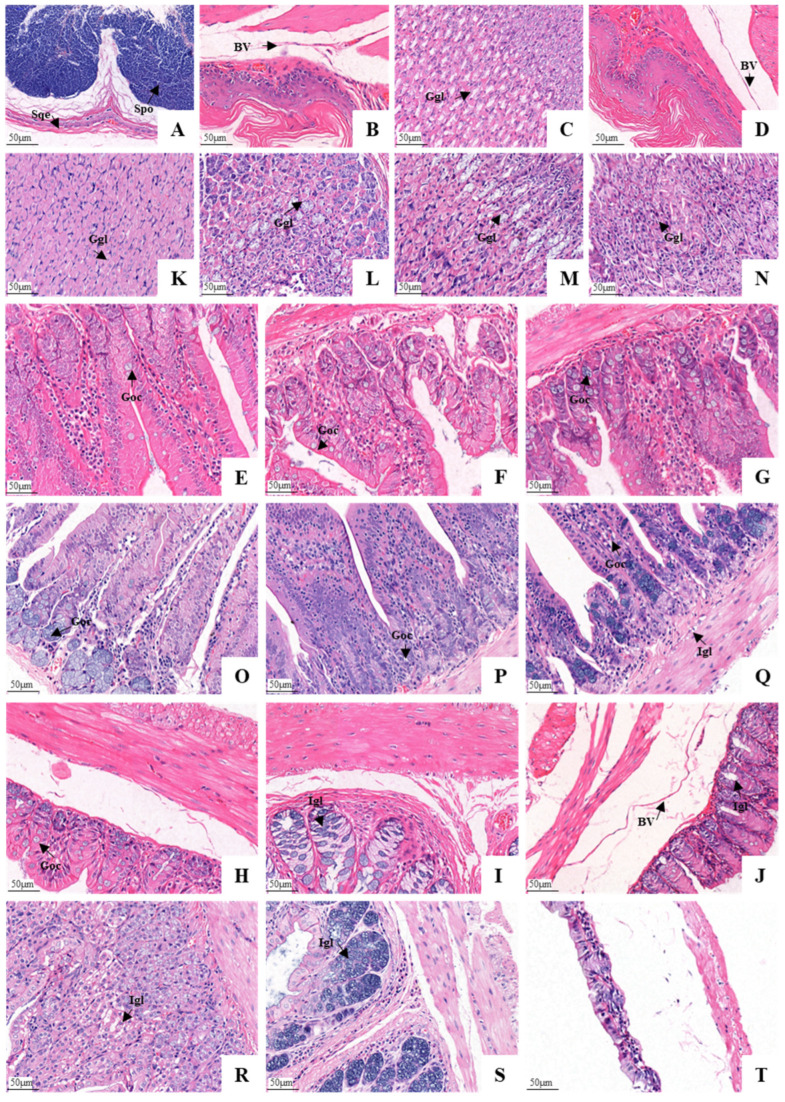
Histological comparison between the plateau zokor’s and the plateau pika’s gastrointestinal organs (400× magnification). (**A**–**J**) Stomach, ((**A**–**D**) fundus, corpus I, corpus II, and pylorus of stomach), duodenum, jejunum, ileum, colon, rectum, and cecum of the plateau zokor, respectively; (**K**–**T**) gastrointestinal organs of the plateau pika, in the same order as above. BV: blood vessels; Goc: goblet cells; Ggl: gastric glands; Igl: intestine glands; Spo: spongy structure; Sqe: squamous epithelium.

**Figure 8 animals-12-02447-f008:**
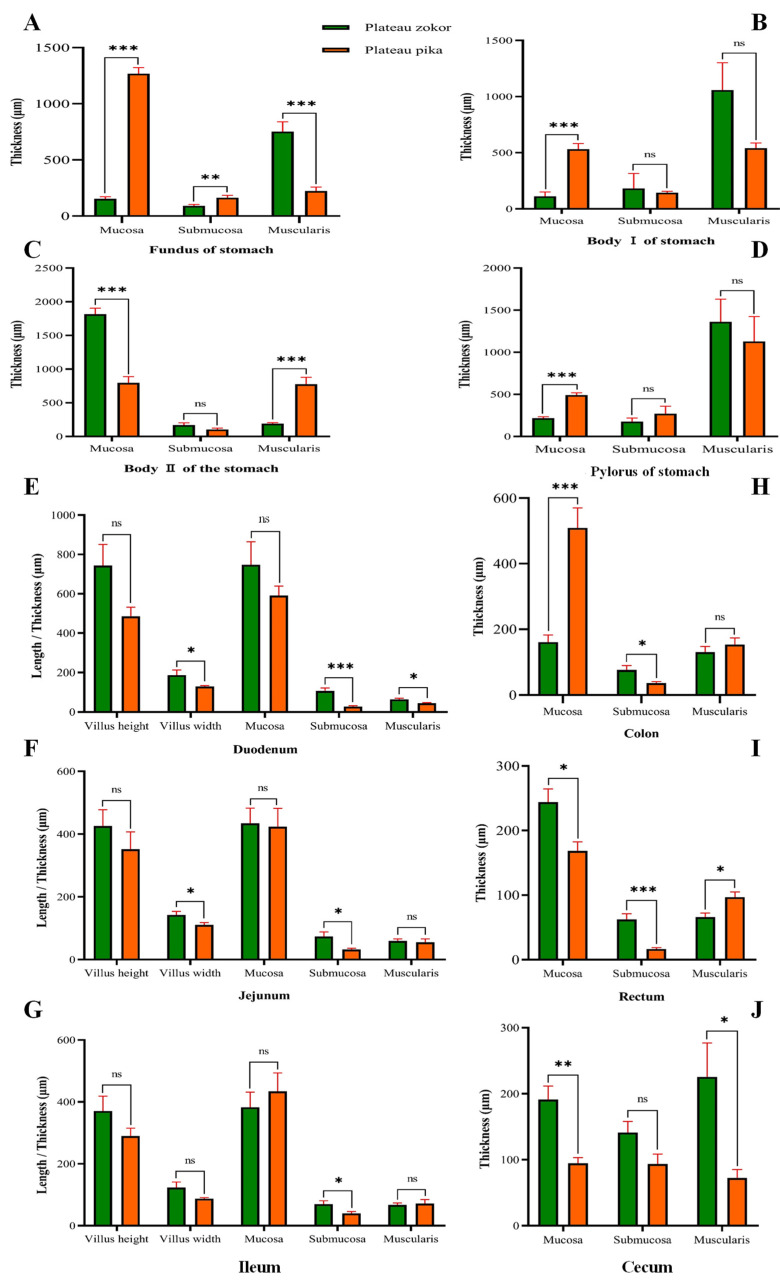
Differences in the histological structure of the various layers of gastrointestinal organs between plateau zokor and plateau pika. (**A**–**J**) Stomach, ((**A**–**D**) fundus, corpus I, corpus II, and pylorus of stomach), duodenum, jejunum, ileum, colon, rectum, and cecum, respectively. * indicates significant difference (*p* < 0.05); ** indicates an extremely significant difference (*p* < 0.01); *** indicates an extremely significant difference (*p* < 0.001); ns indicates an insignificant difference.

**Figure 9 animals-12-02447-f009:**
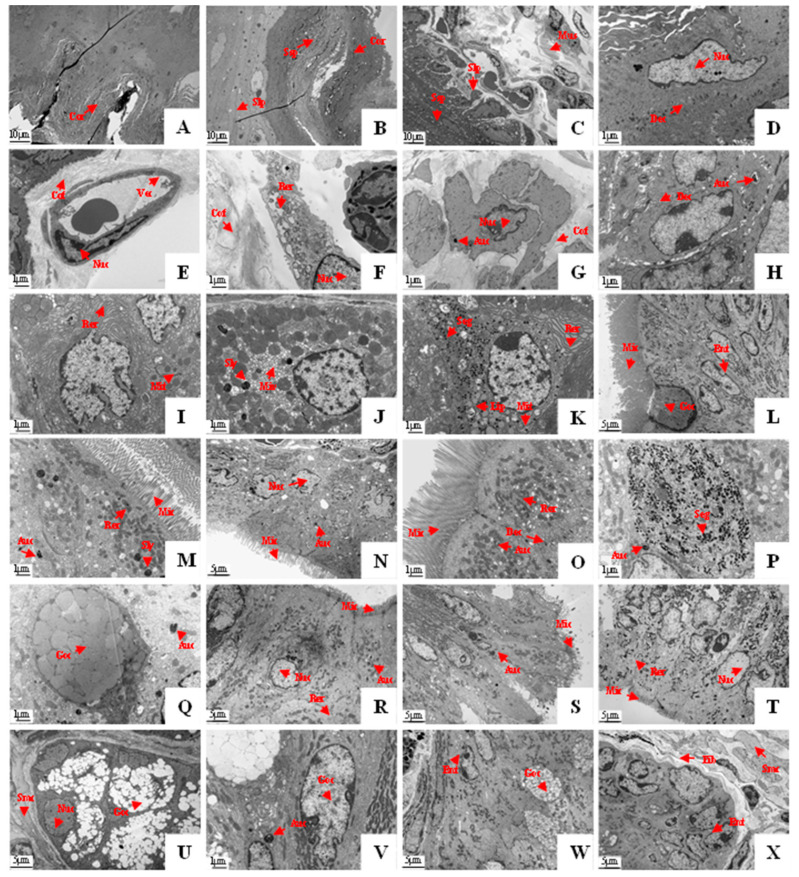
Ultrastructure of the gastrointestinal organs of the plateau zokor ((**A**–**C**) 2400× magnification; (**D**–**K**,**M**,**O**–**Q**,**V**) 12,000× magnification; (**L**,**N**,**R**–**U**,**W**,**X**) 5000× magnification). (**A**–**C**) The mucosal epithelium of the fundus, pylorus, and corpus I of the stomach; (**D**) the spinous cell of the gastric fundus; (**E**) the vascular endothelial cell of the gastric fundus; (**F**) the fibroblast of the gastric corpus I; (**G**) the smooth muscle cell of the gastric pylorus; (**H**) the squamous epithelial cell of the gastric corpus I; (**I**–**K**) the chief cell, parietal cell, and mucous cell of the gastric corpus II; (**L**) the epithelial cell and goblet cell of the ileum; (**M**–**O**,**R**–**T**) the epithelial cell of the duodenum, jejunum, ileum, colon, rectum, and cecum; (**P**,**Q**,**U**–**W**) the goblet cell of the duodenum, jejunum, colon, rectum, and cecum; (**X**) the submucosal lamina propria of the cecum. Auc: autophagy cells; Cof: collagen fibers; Cor: corneum; Dec: desmosome connections; Ent: enterocyte; Fib: fibroblast; Goc: goblet cells; Lip: lipid droplets; Mic: microvilli; Mit: mitochondria; Miv: microtubule vesicles; Mus: muscularis; Nuc: nucleus; Rer: rough endoplasmic reticulum; Seg: secretory granule; Slp: submucosal lamina propria; Sly: secondary lysosomes; Smc: smooth muscle cell; Ssp: stratum spinosum; Vec: vascular endothelial cell.

**Table 1 animals-12-02447-t001:** Differences in gross GIT between the plateau zokor and plateau pika.

	Body Weight (g)	Weight of Gross GIT (g)	Length of Gross GIT (cm)	Organ Index of Gross GIT (%)	Relative Length of Gross GIT (cm/g)
Plateau zokor (*Eospalax baileyi*)	335.49 ± 18.78 (*n* = 22)	19.30 ± 0.72 (*n* = 15)	205.47 ± 6.29 (*n* = 15)	5.99 ± 0.26 (*n* = 15)	0.64 ± 0.03 (*n* = 15)
Plateau pika (*Ochotona curzoniae*)	155.24 ± 5.41 *n* = 21	14.82 ± 0.45 (*n* = 15)	207.50 ± 2.86 (*n* = 15)	10.24 ± 0.33 (*n* = 15)	1.44 ± 0.04 (*n* = 15)
Significances	***	***	ns	***	***

“*n*” represents the number of animal individuals. *** indicates an extremely significant difference (*p* < 0.001); ns indicates an insignificant difference.

## Data Availability

The datasets used and/or analyzed during the current study are available from the corresponding author on reasonable request.

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
