# Peer review of "The Complex and Well-Developed Morphological and Histological Structures of the Gastrointestinal Tract of the Plateau Zokor Improve Its Digestive Adaptability to High-Fiber Foods"

_animals, 2022, doi:10.3390/ani12182447_

Round 1

Reviewer 1 Report

In general the manuscript is well written with correct description of the obtained results. The materials and methods are described in details. The figures are in appropriate resolution, however the numbering of the figures should be improved. All the misspelling should be also carefully checked.

Author Response

Dear Reviewer #1:

Thank you for constructive suggestions concerning our manuscript entitled “The complex and well-developed morphological and histological structures of the gastrointestinal tract of the plateau zokor improve its digestive adaptability to high-fiber foods.” (ID: animals-1882355), which we submitted to “Animals”. Those comments are all valuable in improving our manuscript, and we have studied the comments carefully and made revisions accordingly which we hope to meet with your approval.

According to your suggestions, we revised the whole manuscript seriously, including modifying all spelling mistakes and improving the figure number and figure titles of the results section, hoping that the revised manuscript can meet with your approval. The revisions are marked in red on the paper. Attached please find the modified original version with track changes and an updated clean version, which we would like to submit for your consideration.

Reviewer 2 Report

This is the review on the manuscript entitled “The complex and well-developed morphological and histological structures of the gastrointestinal tract of the plateau zokor improve its digestive adaptability to digest high-fiber foods”. From the point of view of the article, the logic is clear, with a certain degree of literature collation ability and independent scientific research ability. The workload of this article is moderate and has certain application value. After reading the text carefully, I have the following comments.

Comments:

1.     Lines 79-81: “The plateau zokor…grassland ecosystem.” Please cite the recommended reference here:

Animals 2022, 12, 725. https:// doi.org/10.3390/ani12060725

2.     Line 100: delete “(a subterranean rodent)” since you have demonstrated it clearly in the paragraph 3.

3.     Lines 114-115: change “(nine male and thirteen female)” into (nine males and thirteen females); “(eleven male and ten female)” into (eleven males and ten females)

4.     Line 116-117: What are the criteria for determining whether a captured animal is an adult? The criteria for plateau zokor and plateau pika is different, please clear it.

5.     Lines 118-119: male-males; female-females

6.     Line 137: “2.2 Experimental design” needs to be italic.

7.     Lines 145-146: add reference here.

8.     Line 180: How to distinguish duodenum, jejunum, and ileum? There is no difference at all in Figure 1. What is the basis for distinguishing them? Please elaborate here.

9.     Line 181-186: add reference for the method here.

10. Lines 186-187: “Microphotographs…determination”. What is the magnification? The same magnification for each tissue?

11. Lines 191-197: add reference for the method here.

12. Line 223: delete “the same as below”, and add the meaning of each symbol in the title of each figure.

13. Lines 253-256: “The results showed that the organ index of each part of the GIT (stomach, duodenum, jejunum, ileum, colon, rectum and cecum) and the gross gastrointestinal organ index of the plateau zokor were extremely significantly lower than those of the plateau pika (P<0.001) (Figure 5A, Table 1).” In figure 5A, where is the organ index of duodenum? where is the organ index of jejunum? where is the organ index of ileum? There is only the organ index of small intestine, large intestine, and cecum. What's the point of dividing A, B and C into smaller parts? (2) What do you mean by “the gross gastrointestinal organ index”? In table 1, it was described as “Organ index of gross GIT”, please unify.

14.  Figure 5, Table 1, Figure 8: Plateau zokor and plateau pika are two completely different species. When comparing the adaptive evolution of organ phenotypes between different species, interspecific morphological differences have a great impact on the results. How did you eliminate the effects of interspecies differences?

15. Figure 6, 7 and 9: (1) What is the magnification? Add it in the figure title; (2) delete “the same as below” in the figure title; (3) please add arrows after the initials of each section in the figure to indicate the position of each section more clearly

16. Figure 6 and 7: The present alphabetical arrangement of figures is not easy to understand. Please use the capital letters A-J for plateau and lowercase letters a-j for plateau pika. It will be a little bit more intuitive.

17. Figure 8E: The position of one “ns” marked in the figure is not appropriate.

18. Lines 422-425: “Therefore, we suggest that…this adaptability well.” As mentioned above, when comparing the adaptive evolution of organ phenotypes between different species, interspecific morphological differences have a great impact on the results. I think this is limitation of the present study and should at least be mentioned in the discussion.

19. Please check the normal distribution and homogeneity of variance of your original data.

Author Response

Dear Reviewer #2:

Thank you for constructive suggestions concerning our manuscript entitled “The complex and well-developed morphological and histological structures of the gastrointestinal tract of the plateau zokor improve its digestive adaptability to high-fiber foods.” (ID: animals-1882355), which we submitted to “Animals”. Those comments are all valuable in improving our manuscript, as well as the important guiding significance to our research. We have studied the comments carefully and made revisions accordingly which we hope to meet with your approval.

According to your suggestions, we revised the whole manuscript seriously, including the English language style and all spelling mistakes. In addition, we added the relevant references, improved the figure and figure titles of the results section, and modified the corresponding position of the methods and discussion sections. We hope that the revised manuscript can meet with your approval. The revisions are marked in red on the paper. Attached please find the modified original version with track changes and an updated clean version, which we would like to submit for your consideration. The comments you gave me, please see below for my revised description.

Major suggestions/concerns

Questions 1: Lines 79-81: “The plateau zokor…grassland ecosystem.” Please cite the recommended reference here: Animals 2022, 12, 725. https:// doi.org/10.3390/ani12060725

Response: Revised (see line 83 in updated version). Thank you for your comment. We have cited the reference that “[15] An K, Yao B, Kang Y, Bao M, Tan Y, Pu Q, Su J. Seasonal Expression of Gonadotropin Genes in the Pituitary and Testes of Male Plateau Zokor (Eospalax baileyi). Animals. 2022; 12(6):725.”

Questions 2:  Line 100: delete “(a subterranean rodent)” since you have demonstrated it clearly in the paragraph 3.

Response: Revised (see line 102 in updated version). We have deleted “(a subterranean rodent)”

Questions 3: Lines 114-115: change “(nine male and thirteen female)” into (nine males and thirteen females); “(eleven male and ten female)” into (eleven males and ten females)

Response: Revised (see lines 116-117 in updated version). Thank you for your comment. We have changed the “male and female” into “males and females”

Questions 4: Line 116-117: What are the criteria for determining whether a captured animal is an adult? The criteria for plateau zokor and plateau pika is different, please clear it.

Response: Revised (see lines 119-121 in updated version). Determining the age of wild animals, especially rodents, is a challenging scientific problem. Many scholars have studied the age structure division of plateau pika and plateau zokor respectively. Based on the correlation between body weight, body length, coat color and age of plateau zokor and plateau pika, they concluded that it is of great significance for plateau pika and plateau zokor to divide their age according to body weight in field experiments. In this study, the body weight of plateau zokors was 335.49±18.78 g, and the body weight of plateau pikas was 155.24±5.41 g. In terms of weight, all captured animals were adults. This we have been clear in the paper.

Questions 5: Lines 118-119: male-males; female-females

Response: Revised (see lines 122-123 in updated version). Thank you for your comment. We have changed the “male and female” into “males and females”.

Questions 6:  Line 137: “2.2 Experimental design” needs to be italic.

Response: Revised (see line 140 in updated version). We have changed “2.2 Experimental design” to italics.

Questions 7: Lines 145-146: add reference here.

Response: Revised (see line 149 in updated version). Thank you for your comment. We have cited the reference that “[12] Chu, B.; Ji, C.P.; Zhou, J.W.; Zhou, Y.S.; Hua, L.M. Why does the plateau zokor (Myospalax fontanieri: Rodentia: Spalacidae) move on the ground in summer in the eastern Qilian Mountains? J. Mammal. 2021, 102, 346-357.”

Questions 8: How to distinguish duodenum, jejunum, and ileum? There is no difference at all in Figure 1. What is the basis for distinguishing them? Please elaborate here.

Response: Revised (see lines 183-185 in updated version). The duodenum, jejunum and ileum in the small intestine are not easy to distinguish. In this study, we only excised and sectioned into small tissue fragments (5 mm) from the duodenum (4 cm from the pylorus), jejunum (one-half distance from the small intestine) and ileum (4 cm from the cecum). This we have been clear in the paper.

Questions 9: Line 181-186: add reference for the method here.

Response: Revised (see line 190 in updated version). Thank you for your comment. We have cited the reference that “[36] Yang, L.Z.; Fang, J.; Peng, X.; Cui, H.M.; He, M.; Zuo, Z.C.; Zhou, Y.; Yang, Z.Z. Study on the morphology, histology and enzymatic activity of the digestive tract of Gymnocypris eckloni Herzenstein. Fish Physiol Biochem, 2017, 43, 1175-1185.”

Questions 10: Lines 186-187: “Microphotographs…determination”. What is the magnification? The same magnification for each tissue?

Response: Revised (see lines 294, 315, 383-384 in updated version). Thank you for your comment. We have marked the magnification of the micrograph in the figure title of the article.

Questions 11: Lines 191-197: add reference for the method here.

Response: Revised (see line 200 in updated version). We have cited the reference that “[36] Yang, L.Z.; Fang, J.; Peng, X.; Cui, H.M.; He, M.; Zuo, Z.C.; Zhou, Y.; Yang, Z.Z. Study on the morphology, histology and enzymatic activity of the digestive tract of Gymnocypris eckloni Herzenstein. Fish Physiol Biochem, 2017, 43, 1175-1185.”

Questions 12: Line 223: delete “the same as below”, and add the meaning of each symbol in the title of each figure.

Response: Revised (see the titles of figure 2, 3, 5, 8 and table 1 in updated version). Thank you for your comment. We have deleted “the same as bellow”, and add the meaning of each symbol in the title of each figure.

Questions 13: Lines 253-256: “The results showed that the organ index of each part of the GIT (stomach, duodenum, jejunum, ileum, colon, rectum and cecum) and the gross gastrointestinal organ index of the plateau zokor were extremely significantly lower than those of the plateau pika (P<0.001) (Figure 5A, Table 1).” In figure 5A, where is the organ index of duodenum? where is the organ index of jejunum? where is the organ index of ileum? There is only the organ index of small intestine, large intestine, and cecum. What's the point of dividing A, B and C into smaller parts? (2) What do you mean by “the gross gastrointestinal organ index”? In table 1, it was described as “Organ index of gross GIT”, please unify.

Response: Revised (see lines 261-265 in updated version). Thank you for your comment. The duodenum, jejunum and ileum in the small intestine are not easy to distinguish. In this study, we only measured the morphological indicators of the large intestine, small intestine and cecum. The duodenum, jejunum and ileum in this results are due to our mistakes, and we have modified them. “The gross gastrointestinal organ index” is non-standard writing, and we have unified it.

Questions 14:  Figure 5, Table 1, Figure 8: Plateau zokor and plateau pika are two completely different species. When comparing the adaptive evolution of organ phenotypes between different species, interspecific morphological differences have a great impact on the results. How did you eliminate the effects of interspecies differences?

Response: Revised (see lines 172-176 in updated version). We agree with your comment. To allow for the effects of individual body size on GIT morphology between different animals, the gastrointestinal organ weight and length were calculated per body weight as the gastrointestinal organ index and relative length of GIT, respectively.

Questions 15: Figure 6, 7 and 9: (1) What is the magnification? Add it in the figure title; (2) delete “the same as below” in the figure title; (3) please add arrows after the initials of each section in the figure to indicate the position of each section more clearly

Response: Revised (see the figure 6, 7, 9 in updated version). Thank you for your comment. We have added the magnification in the figure title. Secondly, we deleted “the same as below”, and added the arrows in each figure correspondently. 

Questions 16: Figure 6 and 7: The present alphabetical arrangement of figures is not easy to understand. Please use the capital letters A-J for plateau and lowercase letters a-j for plateau pika. It will be a little bit more intuitive.

Response: Revised (see figure 6-7 in updated version). Thank you for your comment. We have used the capital letters A-J for plateau and lowercase letters a-j for plateau pika.

Questions 17: Figure 8E: The position of one “ns” marked in the figure is not appropriate.

Response: Revised (see figure 8E in updated version). We have modified the mistakes in figure 8E.

Questions 18: Lines 422-425: “Therefore, we suggest that…this adaptability well.” As mentioned above, when comparing the adaptive evolution of organ phenotypes between different species, interspecific morphological differences have a great impact on the results. I think this is limitation of the present study and should at least be mentioned in the discussion.

Response: Revised (see lines 435-439 in updated version). We agree with your comment. We acknowledge that the result may have interspecific differences in the phenotypic morphology of the organs between different species. And we have declared this in the corresponding part of the discussion.

Questions 19: Please check the normal distribution and homogeneity of variance of your original data.

Response: Revised (see lines 206-208 in updated version). We agree with your comment. The Kolmogorov-Smirnov test and Levene test were used to determine the normal distribution and homogeneity of variance of the original data before proceeding to further statistical analysis. We have declared this in this paper.
